# Evaluation of the Effect of a New Skin Fixation Technique to Avoid Shrinkage of Skin Samples Obtained from Canine Cadavers

**DOI:** 10.3390/ani14192791

**Published:** 2024-09-26

**Authors:** Ligita Zorgevica-Pockevica, Nataliia Kuzhel, Sigita Kerziene, Simona Vincenti

**Affiliations:** 1Division of Small Animal Surgery, Department of Clinical Veterinary Medicine, Vetsuisse Faculty, University of Bern, 3012 Bern, Switzerland; nkuzhelq@gmail.com (N.K.); simona.vincenti@unibe.ch (S.V.); 2Department of Physics, Mathematics and Biophysics, Faculty of Medicine, Lithuanian University of Health Sciences, 47181 Kaunas, Lithuania; sigita.kerziene@lsmu.lt

**Keywords:** dogs, histological tumour-free margin, skin shrinkage, surgical margins, tumour surgery

## Abstract

**Simple Summary:**

The complete resection of tumours is essential for curative surgical treatment in small animal oncology. The histopathological examination of the surgical margins of resected tumours reveals a histological tumour-free margin, which is the gold standard for predicting the recurrence of resected skin tumours. Skin shrinkage, tissue deformation and the folding of the specimen are artefacts associated with the excision and fixation of a specimen that can lead to dramatic changes in histologic tissue dimensions and consequently the size of the surgical margin. To reduce these artefacts, a new technique was investigated. With this method, the shrinkage of the sample in formalin is drastically reduced and the deformation of the specimen is completely eliminated. We believe that this method could improve the diagnostic accuracy of surgical margins and thus reduce the number of false-positive or false-negative histological tumour-free margins.

**Abstract:**

Skin shrinkage begins immediately after surgical incision and is an artefact associated with the excision and fixation of a specimen. Skin shrinkage results in important changes in histologic tissue dimensions and can affect the correct quantification of the histologic tumour-free margin (HTFM). Bilateral and symmetrical circular skin samples with a diameter of 60 mm were taken from the lateral thoracic, flank and femoral regions of dog cadavers, with the samples from one side belonging to the study group and the samples from the same animal from the other side belonging to the control group. The radius and diameter of the specimen were measured immediately after the excision and 10 min later for each sample. The measurements of the study group were taken again after manual re-extension and fixation on a cork plate before formalin fixation and 48 h after formalin fixation. A total of 66 (33 study and 33 control group) samples were collected from 11 canine cadavers. The mean diameter shrinkage after formalin fixation was 18.24% for the control group and 0.64% for the study group. A statistically significant difference between the study and the control group was found (*p* < 0.001). This method of specimen fixation in the study group avoided skin shrinkage and deformation of the specimen in formalin, which we believe improves the diagnostic accuracy of surgical margins and, thus, reduces the number of false-positive or false-negative HTFM.

## 1. Introduction

Surgical excision is one of the essential components of solid tumour treatment in small animal oncology. To achieve curative surgical treatment, complete tumour resection is essential [1,2,3,4,5,6]. A histopathological examination of surgical margins is a crucial prognostic tool to assess the likelihood of recurrence, with complete excision being essential to prevent recurrence [1,3,4,6,7,8,9,10,11]. Depending on the histological findings, the surgical margins can also be defined as positive or negative in relation to the presence of neoplastic cells. Specifically, a negative margin has no neoplastic cells, while a positive margin has neoplastic cells extending to the surgical incision. A positive surgical margin is generally considered an indication to suggest additional treatment methods [4,6]. In addition, resected tissue undergoes significant structural changes immediately after the resection, during the fixation of the material and the preparation for histologic assessment, which may cause artefacts during histologic evaluation [12,13,14]. The physical reduction in tissue length, also called “shrinkage”, occurs immediately when the tissue is detached from the surrounding structures due to the contractility of the myofibrils and the elasticity of the tissue [8,15,16,17,18,19]. After the resection of the specimen and fixation in 10% formalin for 24 h, an average canine skin shrinkage of 15.6% (mean ± SE total diameter change of 6.2 ± 0.7 mm) of the resected tissue was described [8,14]. Skin shrinkage, tissue deformation and the folding of the specimen are artefacts associated with excision and the fixation of the specimen which can result in dramatic changes in histologic tissue dimensions and, consequently, of surgical margin dimensions [13,20]. The distortion of different skin layers (fat, subcutaneous, muscle, and tumour tissue) during fixation and processing can lead to both false-positive and false-negative classifications of the true status of the surgical margin [3,20]. A surgical margin can be defined as “false-negative” when a margin is histologically defined as negative in terms of the presence of neoplastic cells yet neoplastic cells are actually present but are not detected due to the selection of an unrepresentative margin area or in association with ink and sectioning artefacts. This is the consequence of a technical limitation related to the trimming procedure, when the different layers of the specimen are in a non-anatomical position due to the shrinkage and deformation of the specimen during fixation. A false-negative HTFM may lead to an incorrect prognosis and possibly unexpected tumour recurrence [4]. In human studies, the incidence of local recurrence following cancer resection with free margins varied between 4% and 32% [21], and in rectal cancer a local recurrence of 10% was described in HTFM [22]. 

If the surgical margins are histologically described as positive, but no residual neoplastic cells are present, this is referred to as a “false positive” surgical margin. A false-close HTFM or a false-positive HTFM can be associated with tissue shrinkage, deformation and the mishandling of the specimens after resecting and during fixation in formalin [23]. It can lead to unnecessary re-excision or unnecessary additional therapy (chemotherapy or radiotherapy), which has a high potential to cause additional costs and may harm the animal [24]. The aim of our study was to develop a new, easy-to-use and cost-effective method to prevent the occurrence of histologic processing artefacts which are associated with the excision and fixation of the specimen, which would allow for the obtainment of more accurate information about the surgical margins. We hypothesised that this technique could dramatically reduce the percentage of skin shrinkage experienced by the skin samples after surgical excision.

## 2. Materials and Methods

Eleven different breeds of dog cadavers were included in the study. All cadavers were donated for research purposes and were euthanised for other reasons than tumours or skin diseases. The samples were collected 3–8 h after euthanasia, and 3 cadavers after the euthanasia were immediately frozen and then thawed for the study.

The Body Condition Score (BCS) was assessed from 1 to 9 according to the Nestle Purina Body Condition Score table.

The skin samples, including the subcutis and underlying fascia, were taken bilaterally and symmetrically from the thorax (T), flank (F) and upper thigh (UT) regions. 

The first region was the lateral side of the thorax immediately caudal to the forelimbs. The samples were taken from the flank region immediately caudal to the rib cage. The samples from the upper thigh region were taken from the hind leg at approximately mid-thigh. 

The samples were circular in shape and the pre-excisional size of the samples was 60 mm in diameter. Premade flexible plastic templates with a diameter of 60 mm were used to mark the excision areas with skin markers. The centre was marked in the middle of the drawn radius, and the radius was divided with vertical and horizontal lines. Cranial, caudal, dorsal and ventral directions on the samples were marked (Figure 1A).

Sampling was always started from the left side. The samples were assigned to the study or control group by tossing a coin.

A skin incision was made along the drawn line through the skin, subcutis, and superficial fascia (Figure 1B). The dorsal direction of the samples was marked with a single interrupted suture pattern suture, and the cranial direction with two simple interrupted suture pattern sutures. The deep margin was dissected with scissors, and the specimen collection was completed. The lateral and deep margins of the specimens were stained with ink after resection (Figure 2A).

The measurements were carried out immediately after the sample was taken (Figure 1C). The measurements were taken with an accuracy of 0.01 mm using a digital calliper. First, the radius of the sample was measured in four directions.

The first measurement was from the centre to the dorsal edge of the specimen and was named C-D; the second measurement was from the centre of the sample in the caudal direction and was named C-Ca. The third measurement was from the centre to the ventral edge of the sample and was named C-V. The last measurement was from the centre to the cranial edge of the sample and was named C-Cr. Subsequently, the diameters of the samples in the dorsoventral (DV) and craniocaudal (CrCa) directions were measured. The measurement of the radius and diameters was repeated after ten minutes in all directions (Figure 2C).

A radius of 60 mm was drawn on a 3 mm thick piece of cork board (Figure 2B); the excised samples were extended manually and then fixed in place with pinpoint needles (Figure 3A,B), and the previous measurements were repeated.

The samples were fixed in 10% neutral buffered formalin for at least 48 h. The previously described measurements (radius and diameter) were repeated after 48 h.

For the control group, the samples were collected in the same way, but the samples were not fixed on a cork board and were instead simply immersed in the formalin solution. 

Specimen resection, fixation and all measurements were performed by a senior resident in a small animal surgery which was supervised by a boarded surgeon for the first few cadavers.

The diameter of the shrinkage of the deep surgical margins was measured in the DV and CrCa directions. The deformation of the specimens was evaluated and documented. 

Statistical analysis was performed with SPSS software (version 29.0.0.0 (241), IBM Corp. licence 1989, 2022; New York, NY, USA). The samples were small (n = 22), and the Shapiro–Wilk test confirmed that not all data were normally distributed. Dimensional differences between the study and control groups were expressed by calculating the means and standard errors of the means. The statistical significance of the differences between the groups was determined using Student’s *t*-test for dependent samples. The dimensional differences between the study and control groups were expressed by calculating the medians and limits. The statistical significance of the differences between the groups was determined using Wilcoxon Signed Ranks Test for dependent samples. The data were considered statistically significant when *p* < 0.05.

## 3. Results

A total of 66 samples were collected from three different regions (thorax, flank, upper thigh) from each side of 11 cadavers, with 33 samples included in the study and another 33 samples included in the control group. In the study, five of the dogs were mixed-breed dogs, and six dogs were different breeds (Jack Russell terrier, pug, Labrador retriever, French bulldog, Shar Pei and Bernese Mountain dog). The median body weight was 17.05 kg and ranged from 6.7 kg to 35.6 kg. The average age of the animals was 8.6 years with a range of 5.4 to 14.0 years. 

When the animals’ body conditions were assessed, it was found that seven dogs’ cadavers were in an ideal condition (BCS 4–5/9); one dog was underweight (BCS 3/9), and three animals were obese (BCS 6–7/9).

The median specimen radius shrinkage of different regions immediately after specimen resection, ten minutes after resection and 48 h after fixation in formalin is shown in Appendix A. 

The median diameter with changes in the specimens is shown in Table 1. The changes were studied in the following time frames: immediately after the resection of the specimens, ten minutes after the resection and 48 h after fixation in formalin.

The mean percentage of skin shrinkage of the specimens in different regions immediately after the resection of the specimens, ten minutes after the resection and 48 h after the fixation in formalin is shown in Figure 4 and Appendix A.

The mean percentage of skin shrinkage immediately after the excision ranged from 11.30% ± 1.96 (SE) (T-DV) to 15.51% ± 1.53 (SE) (UT-DV) in the different regions of the control group. 

The mean percentage of skin shrinkage immediately after the excision of the specimens in the study group varied from 6.66% ± 1.28 (SE) (T-DV) to 20.12% ± 2.10 (SE) (F-CrCa, *p* < 0.01). 

The mean percentage of skin shrinkage 10 min after the excision ranged from 13.77% ± 2.68 (SE) (T-CrCa) to 20.71% ± 1.60 (SE) (UT-DV) in the different regions of the control group.

The mean percentage of skin shrinkage 10 min after the excision of the specimens in the study group ranged from 15.21% ± 2.51 (SE) (UT -CrCa) to 19.7% ± 1.64 (SE) (UT-DV). 

After stretching and the fixation on the cork plate, the mean diameter in millimetres of the specimens in the study group in the different regions ranged from 61.22 ± 0.73 (SE) mm (F-DV) to 62.55 ± 0.55 (SE) mm (T-CrCa). 

The mean percentage of skin shrinkage of the control group specimens 48 h after 10% formalin fixation ranged from 16.53% ± 1.63 (SE) (F-CrCa) to 21.73% ± 2.72 (SE) (UT-DV). 

The mean skin shrinkage of the specimens from all the regions of the study group after the 48 h fixation in 10% formalin ranged from 0.42% ± 0.73 (SE) (T-DV) to 0.85% ± 0.57 (SE) (T-CrCa). All differences were statistically significant (*p* < 0.001).

The diameter of the deep margins of the study group shrank less after 48 h of fixation in formalin than in the control group, but no statistical significance was found.

Three cadavers were frozen immediately after euthanasia and then thawed. No statistically significant difference was found concerning the amount of skin shrinkage between the fresh and thawed cadavers and among different BCS.

In the control group, distortions between tissue layers (Figure 5B), deformations, the twisting (Figure 5A) of the specimen and skin folding at the surgical margin were observed in all cases but to varying degrees.

In mild cases, the specimen was deformed, but in severe cases, the subcutaneous tissue with deep surgical margins slipped far from the normal anatomical position. Figure 5C shows the three-dimensional deformations of the specimen found in the control group. No signs of deformation or twisting of the specimen were found in the study group.

## 4. Discussion

The findings of the current study have allowed us to validate our hypothesis. Indeed, through our straightforward and cost-effective method of tissue fixation on a cork plate, we were able to significantly decrease the rate of skin shrinkage to just 0.64%. In contrast, specimens in the control group experienced a shrinkage rate of 19%. To date, no formalin fixation method has been described that allows the diameter of a macroscopically intact skin specimen to remain approximately the same size as before resection.

In this study, the shrinkage of the specimens immediately after resection was observed in 13.78% and 10 min after resection in 16.71% of the specimens. Despite the above-mentioned shrinkage of the sample, the samples of the control group could be fixed on the cork plate using a new method for fixing the skin sample at the size before excision. Previous studies have described physical tissue shrinkage immediately after toe excision due to myofibril contractility and tissue elasticity [12,14,16,18,19,20,25,26]. The immediate change in specimen length and the reduction in width in humans showed a 12% reduction in specimen length and a 9% reduction in width [25].

In this study, no statistically significant difference in tissue shrinkage was observed between the three different regions, but it is worth noting that the highest percentage of specimen shrinkage was observed in the upper thigh region ten minutes after specimen resection (UT-DV), where tissue shrinkage reached 21.71% in the control group and 19.70% in the study group. In the most comprehensive veterinary study, skin shrinkage was examined in 216 skin samples from different anatomical regions. Significant correlations were not found between the percentage of skin shrinkage and the side or region of the body or the orientation of the measurements in relation to the skin tension lines [14,20]. In another veterinary study, it was described that the samples from the hind limbs showed a significantly (*p* < 0.001) greater decrease in length than the samples from the lumbar region and the head [27]. Miller and Dark described that tissue shrinkage after the fixation of the specimen in formalin did not differ from the abdomen or hind limbs and was not statistically significant; only specimen shrinkage from the thoracic region was statistically significant, so tissue shrinkage was described [12]. In human studies, tissue shrinkage due to fixation in formalin was found to be 8% in length and 2.6% in width [25], but in another study, the effects of formalin on tissue shrinkage were denied [28]. In contrast to the aforementioned study, in the present study, the shrinkage of the skin was observed in the control group, and minimal shrinkage was observed in the study group, despite the sample being fixed to the cork plate with pinpoint needles.

In a cat skin shrinkage study, skin and fascia were fixed using the simple continuous suture pattern and the “four-quadrant suture pattern”. No statistically significant difference was found between the skin shrinkage of the sutured and non-sutured specimens [19].

In our study, different degrees of tissue shrinkage were observed in the different layers of the specimen. The diameter of the deep surgical margins was significantly different from the diameter of the skin layer in both the study and control groups; the deep surgical margins shrank more compared to the skin and subcutaneous layers. Different tissue structures may exhibit different tissue shrinkage in formalin. The tissue shrinkage of canine small intestine specimens after the fixation of the specimen in formalin for 24 h was 26.3% [26]. Tissue shrinkage in the oral mucosa of the dogs after the initial resection to the final microscopic assessment of the mucosal margins on the tongue surface was 30.7% [29]. Structural changes in the skin may influence skin shrinkage [13,20,30]. Studies of excised cutaneous mast cell tumours with the surgical margins showed that the mean total skin shrinkage of the specimen (surgical margins and tumour tissue) was 17.70%. The mean skin shrinkage within the grossly visible tumour was lower and was only 4.45%, but the surrounding gross normal skin showed a shrinkage of 24.42% [20]. The above study shows that skin shrinkage was higher in macroscopically unaltered skin taken from a living organism than from animal cadavers. In patients with canine mast cell tumours (MCT) and canine soft tissue sarcomas (STS), the shrinkage of the gross normal surgical margin (GNSM) was observed at 80% after the resection of the specimen. The tissue shrinkage of the gross normal surgical margin of MCT specimens after (the) fixation in formalin was 75% (60–90%) and of STS was 87% (60–103%) [30].

In our study, deformations of varying degrees were observed in all specimens of the control group when the samples were fixed in formalin, whereas no deformation of the specimen was observed in the study group. 

These tissue deformations can lead to difficulties in histological evaluation and cause false-positive or false-negative surgical margin status in tumour patients [8]. There was no evidence of specimen deformation or twisting in the study group. The new skin fixation technique eliminates the artefact of sample deformation associated with sample fixation in formalin. It should also reduce the false positive or false negative HFHM associated with sample deformation and distortion.

Resected specimens consisting of different layers during the fixation of the specimens may have different rates of distortion between layers, which may result in a significant deformation of the resected specimen [3,26]. The fixation of the fascia to the skin with the simple continuous suture pattern during specimen collection reduced rotation and translation of the tissue layers but had no significant effect on specimen shrinkage or distortion compared to the non-sutured specimen [19,31].

This is a new yet undescribed method of sample fixation, so it cannot be completely ruled out that there exist new, previously undescribed sample fixation artefacts in connection with this type of sample fixation. Further research is needed to answer this question.

The present study has a few limitations. First, because of the small number of the donated animals in this study, it was not possible to form separate study groups based on the age, sex and breed of the animals. Therefore, in our study, it was not possible to determine how skin shrinkage might be affected by the breed, sex and different ages of the animals. The results of our study reflect skin shrinkage in middle-aged and older animals because the average age of the animals in our study was 8.6 years. In humans, skin shrinkage studies have shown that an increasing patient age and solar elastosis correlate with reduced shrinkage [17].

Furthermore, while our study utilised cadaveric models to investigate the feasibility and initial outcomes of the trimming techniques, we acknowledge that the practical applicability of these findings in a live clinical context requires further validation.

Future research should focus on experimental studies involving live laboratory animals to assess the real-time efficacy and potential challenges of these techniques in a clinical setting.

Additionally, randomisation software should be used to ensure an unbiased sample allocation, and the impact of cellular distortion, skin thickness, and subcutaneous tissue on fixation outcomes should be further explored. Including samples from distal limb areas and applying these techniques to small biopsies could provide valuable comparative data and extend the applicability of the findings of the present study. Combining macroscopic analysis with histopathological examination will offer deeper insights into cell distribution and the effects of shrinkage on microscopic areas. Finally, follow-up studies with larger and more diverse sample sizes will be essential for validating our results and understanding the efficacy of these techniques across different demographics.

Only animals with no history of skin diseases were included in the study, and no skin lesions were detected during the evaluation of body conformation in the study.

The health status of the animals’ skin was assessed macroscopically, but no histological examinations were performed.

## 5. Conclusions

This study shows that after the excision of the specimen and the initial skin shrinkage that occurs immediately after the removal of the specimen, it is possible to stretch the skin to its former diameter and prevent its re-shrinkage by fixing the specimen on a cork plate. The further fixation of the specimen in formalin shows minimal tissue shrinkage. This method eliminates the deformation of the specimen in formalin, which we believe improves the diagnostic accuracy of surgical margins and, thus, reduces the number of false-positive or false-negative HTFM. This method does not require large additional financial resources and is easy to apply in practice. However, difficulties may arise in fixing specimens with a large diameter.

## Figures and Tables

**Figure 1 animals-14-02791-f001:**
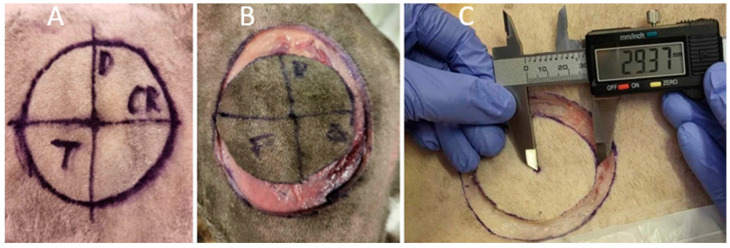
Preparation of the specimen. (**A**) A planned line of incision, 60 mm in diameter with a circular shape, with a marked centre and diameter divided by vertical and horizontal lines and marked directions. (**B**) The lateral margin incision was made along a previously drawn line. (**C**) Measurement of the radius immediately of the lateral margin incision.

**Figure 2 animals-14-02791-f002:**
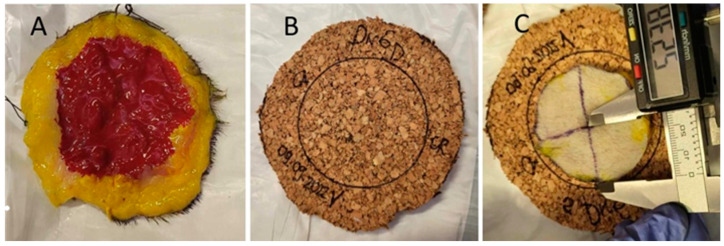
Preparation of the specimen. (**A**) The lateral surgical margin is coloured yellow, and the deep surgical margin is coloured red. (**B**) A radius of 60 mm is drawn on a 3 mm thick piece of cork board to fix the specimen of the study group. (**C**) Radius measurement of the samples of the study group at 10 min after resection.

**Figure 3 animals-14-02791-f003:**
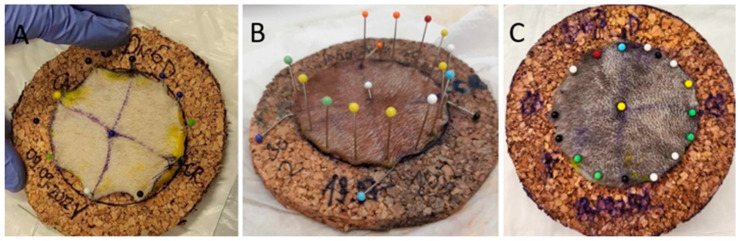
Preparation of the specimen. (**A**) The specimen of the study group is extended and partially attached to the cork board, observing the 60 mm line drawn on it. (**B**) A fully fixed sample of a study group on a cork board. (**C**) Samples of the study group after 48 h fixation in 10% neutral buffered formalin.

**Figure 4 animals-14-02791-f004:**
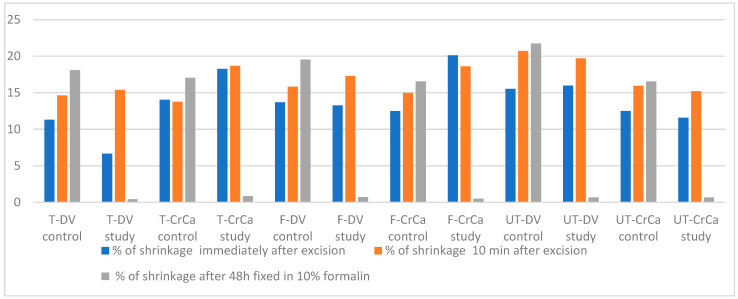
The mean percentage of skin shrinkage of the specimens in different regions immediately after the resection of the specimens, ten minutes after the resection and 48 h after the fixation in formalin.

**Figure 5 animals-14-02791-f005:**
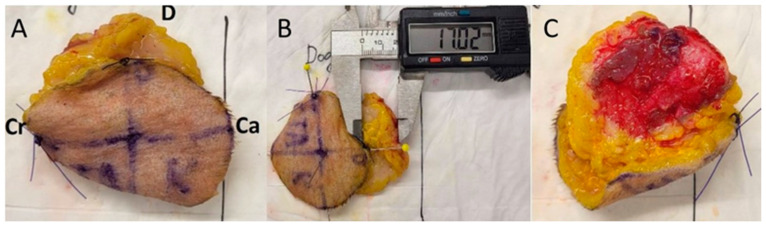
Sample of the control group after 48 h fixation in 10% neutral buffered formalin. (**A**) Severe deformations (twisting) of the specimen (D is the dorsal margin, Ca the caudal and Cr the cranial margin of the specimen). (**B**) Displacement of different tissue layers of the control group specimen. (**C**) Displacement of lateral and deep surgical margin of the control group specimen.

**Table 1 animals-14-02791-t001:** Median diameter (min.–max.) of the specimens in thoracic, flank and upper thigh regions in different time periods.

	Immediately after Excision, mm		10 min after Excision, mm		After 48 h Fixed in 10% Formalin, mm	
	Control	Study	Control	Study	Control	Study
Diameter T-DV	52.8 ^a^ (48.0–59.1)	56.0 ^b^ (52.3–59.8)	52.8 (48.0–59.1)	56.0 (52.3–59.8)	48.6 ^c^ (42.2–59.3)	61.3 ^d^ (56.0–69.0)
Diameter T-CrCa	52.8 (43.8–61.8)	50.9 (42.0–53.0)	52.8 (43.8–61.8)	50.9 (42.0–53.0)	49.1 ^c^ (40.5–58.0)	62.0 ^d^ (58.0–65.6)
Diameter F-DV	53.1 (45.0–55.8)	52.4 (46.0–57.4)	52.1 (41.0–57.0)	48.8 (45.0–55.9)	48.0 ^c^ (38.0–56.6)	60.4 ^d^ (57.0–65.3)
Diameter F-CrCa	53.1 ^c^ (47.9–56.0)	49.0 ^d^ (41.4–53.7)	52.0 (44.6–54.6)	51.0 (40.1–58.0)	50.0 ^c^ (46.4–56.2)	62.3 ^d^ (57.8–64.1)
Diameter UT-DV	50.0 (46.1–58.0)	50.0 (43.7–57.0)	46.7 (44.5–56.0)	47.3 (43.7–54.0)	46.3 ^c^ (41.6–58.0)	61.2 ^d^ (60.2–64.0)
Diameter UT-CrCa	53.0 (46.3–56.0)	53.5 (40.9–62.0)	51.7 (46.2–54.0)	51.5 (40.6–61.0)	51.2 ^c^ (43.4–57.0)	62.3 ^d^ (57.5–65.5)

a,b—*p* < 0.05; c,d—*p* < 0.01 different letters indicate statistically significant differences between control and study groups.

## Data Availability

The data presented in this study are available on request from the corresponding author upon reasonable request.

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
