# Peer review of "Evaluation of the Effect of a New Skin Fixation Technique to Avoid Shrinkage of Skin Samples Obtained from Canine Cadavers"

_animals, 2024, doi:10.3390/ani14192791_

Round 1

Reviewer 1 Report

Comments and Suggestions for Authors

In the current study, the authors set out to determine how the use of cork bands reduces shrinkage of formalin-fixed skin samples for diagnostic purposes. The work is a valuable study and, despite its simple methodology, contributes to the development of veterinary diagnostics contributes to the development of veterinary diagnosticsI consider this contribution to be highly innovative and with great potential for application.

The subject of the study is properly discussed in the light of previous findings and current knowledge.

Minor comments:

Line 79 – subcutis histologically is a part of skin.

Line 80 - please use the correct anatomical names of the area from which the skin samples were taken.

Line 162 – p was <0.01 or <0.05 (line 144)?

Author Response

Thank you for the review! See the attached document for details.

Reviewer 2 Report

Comments and Suggestions for Authors

Dear Authors,

In my opinion, this attractive, well-written article does not have any significant errors. It describes an interesting method of preserving histological samples to eliminate potential errors in assessment by a histopathologist. However, this is a method that an oncological surgeon should use. The surgeon or someone from his team preserves the material in formalin to send it for histopathological examination. There are few veterinary histopathology laboratories; much of the material is sent to them as courier shipments. Only in academic centers is the surgery clinic adjacent to the pathology laboratory.

I do not understand the description for Figure 2—it concerns color markings, and the photos are black and white. In item 7 of the literature, the first author's name is incorrect. It should be Kisser PK, not Iser PK. Please correct.

Best regards

Author Response

Thank you for the review! Please refer to the attached document for details.

Reviewer 3 Report

Comments and Suggestions for Authors

Keywords - double "skin shrinkage"

Introduction:

Line 54 - [8,9,] - should be corrected

The introduction is concise and clear. I think the concept of false positive and negative margins should be extended. 

Materials and Methods

It is well written. There is quite small number of cadavers, but Authors are aware.

Results

Line 158 - after (the) fixation - should be corrected

Information on the condition of the dogs should be added. As stated in the introduction - the amount of body fat can make a difference to fixation. Please add this in the manuscript. However, the freezing process itself causes changes in tissue parameters. Therefore, it should be indicated which sections are from frozen tissue. Maybe also do a statistical comparison, will there be other parameters for shrinkage, tissue elasticity?

Discussion

It is well prepared.

References

Should be corrected according to the guidance.

Author Response

(The authors gave the same response as above.)
